# The chromatin-remodeling enzyme Smarca5 regulates erythrocyte aggregation via Keap1-Nrf2 signaling

Yanyan Ding[1,2,3,4†], Yuzhe Li[5,6†], Ziqian Zhao[2,3,4†], Qiangfeng Cliff Zhang[5,7], Feng Liu[2,3,4*]

[1]The Max-Planck Center for Tissue Stem Cell Research and Regenerative Medicine, Bioland Laboratory, Guangzhou, China; [2]State Key Laboratory of Membrane Biology, Institute of Zoology, Chinese Academy of Sciences, Beijing, China; [3]Institute for Stem Cell and Regeneration, Chinese Academy of Sciences, Beijing, China; [4]University of Chinese Academy of Sciences, Beijing, China; [5]MOE Key Laboratory of Bioinformatics, Beijing Advanced Innovation Center for Structural Biology & Frontier Research Center for Biological Structure, Center for Synthetic and Systems Biology, School of Life Sciences, Tsinghua University, Beijing, China; [6]Academy for Advanced Interdisciplinary Studies, Peking University, Beijing, China; [7]Tsinghua-Peking Center for Life Sciences, Beijing, China

*For correspondence:
liuf@ioz.ac.cn

†These authors contributed equally to this work

Competing interest: The authors declare that no competing interests exist.

**Abstract** Although thrombosis has been extensively studied using various animal models, our understanding of the underlying mechanism remains elusive. Here, using zebrafish model, we demonstrated that *smarca5*-deficient red blood cells (RBCs) formed blood clots in the caudal vein plexus. We further used the anti-thrombosis drugs to treat *smarca5*[zko1049a] embryos and found that a thrombin inhibitor, argatroban, partially prevented blood clot formation in *smarca5*[zko1049a]. To explore the regulatory mechanism of *smarca5* in RBC homeostasis, we profiled the chromatin accessibility landscape and transcriptome features in RBCs from *smarca5*[zko1049a] and their siblings and found that both the chromatin accessibility at the *keap1a* promoter and expression of *keap1a* were decreased. Keap1 is a suppressor protein of Nrf2, which is a major regulator of oxidative responses. We further identified that the expression of *hmox1a*, a downstream target of Keap1-Nrf2 signaling pathway, was markedly increased upon *smarca5* deletion. Importantly, overexpression of *keap1a* or knockdown of *hmox1a* partially rescued the blood clot formation, suggesting that the disrupted Keap1-Nrf2 signaling is responsible for the RBC aggregation in *smarca5* mutants. Together, our study using zebrafish *smarca5* mutants characterizes a novel role for *smarca5* in RBC aggregation, which may provide a new venous thrombosis animal model to support drug screening and pre-clinical therapeutic assessments to treat thrombosis.

## Editor's evaluation

The detailed kinetics and underlying mechanism of thrombosis formation remains elusive. In this study, Liu and colleagues revealed the development of venous thrombosis upon loss of Smarca5 in zebrafish embryos and the Smarca5-mediated epigenetic regulation of oxidative response genes in erythroid cell function. This study not only establishes a new venous thrombosis animal model, but also identify the Smarca5-Keap1-Nrf2-Hmox1 axis as a new regulatory pathway in the pathophysiology of thrombosis.

## Introduction

The erythrocytes, or red blood cells (RBCs), are highly differentiated cells produced during erythropoiesis. Mature RBCs are characterized for their abundance of hemoglobin, which can deliver oxygen to surrounding tissues. Importantly, the flexible structure of RBCs makes it capable of traveling through all blood vessels including capillaries by deformation (*Rodríguez-García et al., 2016*). On the benefit of accumulated hemoglobin and the deformation ability, RBCs are essential for organism development by facilitating tissue oxygen delivery and transporting carbon dioxide into the respiration tissues. Moreover, RBCs participate in the maintenance of thrombosis and hemostasis (*Weisel and Litvinov, 2019*).

Epigenetic regulation of RBC-related genes is fundamental for normal development and maintenance of RBCs (*Hewitt et al., 2014*). In this process, the regulation of chromatin accessibility is a prerequisite for gene transcription and is regulated by chromatin remodelers. For instance, Brg1 could regulate α- and β-globin gene transcription in primitive erythrocytes in mice (*Bultman et al., 2005*; *Griffin et al., 2008*). The nucleosome remodeling and histone deacetylase (NuRD) is identified to activate human adult-type globin gene expression (*Miccio and Blobel, 2010*).

Disorder of the gene regulation in RBCs will lead to cellular defects, thereby causing multiple diseases, such as hemoglobinopathy-induced anemia, RBC lysis-induced hemolytic anemia, and thrombosis (*Kato et al., 2018*; *Roumenina et al., 2016*; *Weisel and Litvinov, 2019*). Among them, thrombosis is a leading cause of death worldwide (*Wendelboe and Raskob, 2016*). In contrast to arterial thrombi, which are rich in platelets, the venous thrombi are enriched in fibrin and RBCs (*Mackman, 2008*; *Mackman et al., 2020*). Moreover, venous thrombi can break off, travel, and lodge in the lung, thereby causing pulmonary embolism (*Wolberg et al., 2015*).

Currently, the ligature-based inferior vena cava models, free radical thrombosis models and genetic knockout models are widely used in mice to study deep vein thrombosis (*Diaz et al., 2019*; *Grover and Mackman, 2019*). These disease models are generated mainly through disrupting blood flow, endothelium and blood coagulability. Taking advantage of the conserved hemostatic system and the transparency of embryos, zebrafish has also been used to generate thrombosis models. For instance, phenylhydrazine-treated zebrafish develop severe thrombosis in the caudal vein (*Zhu et al., 2016*). Mechanistically, phenylhydrazine causes externalization of phosphatidylserine on plasma of RBC membrane and generates oxidative radicals, thereafter, resulting in the thrombosis formation. These studies in animal models shed light on the understanding and treatment of vaso-occlusion phenotype in patients with RBC defects. However, the detailed kinetics and underlying mechanism of thrombosis formation in these models are not fully explored.

In our previous study, genetic deletion of an epigenetic regulator-*smarca5* (*smarca5^{zko1049a}*) resulted in abnormal chromatin accessibility, and we observed disruption of hematopoietic transcription factor binding in the genome, finally leading to defects in fetal hematopoietic stem and progenitor cells (HSPCs) (*Ding et al., 2021*). However, whether the other hematopoietic cell types are regulated by *smarca5* is unknown. Here, we develop a new zebrafish RBC aggregation model with a deletion of *smarca5*, loss of which leads to the formation of blood clots in the caudal vein plexus (CVP). We further present how exactly the change in the subcellular structure of *smarca5*-deficient RBCs occurred using transmission electron microscopy (TEM), and uncovered the disintegration of cristae in mitochondria in RBCs. To explore the regulatory mechanism of *smarca5* in RBC homeostasis, we profiled the chromatin accessibility landscape and transcriptome features by performing Assay for Transposase-Accessible Chromatin with high-throughput sequencing (ATAC-seq) and RNA sequencing (RNA-seq) analyses in RBCs from *smarca5^{zko1049a}* and their siblings. Mechanistically, loss of *smarca5* led to the decreased chromatin accessibility at *keap1a* promoter and thus decreased transcriptional expression of *keap1a*. Keap1 is a suppressor protein of Nrf2, which regulates the expression of oxidative response genes. A downstream target of Keap1-Nrf2, *hmox1a*, showed a markedly increased expression upon *smarca5* deletion. Moreover, overexpression of *keap1a* or knockdown of *hmox1a* partially rescued the blood clot formation, supporting that the disrupted Keap1-Nrf2 signaling in *smarca5* mutants led to the RBC aggregation. Collectively, our *smarca5*-deficient zebrafish model may serve as a new venous thrombosis model for drug screening in clinical therapy.

**eLife digest** After an injury, cells in our blood (called red blood cells) often stick together to form clots to stop us from bleeding and prevent infection. These clots, however, can sometimes develop in veins and arteries, resulting in a condition known as thrombosis. If left untreated, these blockages can be life-threatening and lead to a heart attack or stroke.

To study the physical effects of venous thrombosis and test different treatments, researchers often use animal models. In particular, the transparent embryos of zebrafish, as it easy to see how blood flows through their circulatory system. However, it is difficult to explore the underlying mechanisms that cause red blood cells to aggregate together using these models.

To overcome this, Ding et al. developed a new model for venous thrombosis by deleting the gene for a protein called Smarca5. They found that red blood cells lacking this gene were more likely to clump together in the veins of zebrafish. Further experiments showed that this mutation reduced the activity of the gene for a protein called Keap1a, which suppresses the activity of Nrf2.

Nrf2 switches on a number of genes involved in blood clotting, including the gene for the protein Hmox1a. Ding et al. discovered that increasing the activity of the gene that encodes the Keap1a protein, or decreasing the activity of the gene for Hmox1a, partially stopped red blood cells from sticking together in the zebrafish model.

These findings suggest that the blood clots formed in the zebrafish model are due to the disrupted connection between Keap1a and Nrf2. This model could be used to screen new drugs for treating venous thrombosis. However, further experiments are still needed to see how similar the blood clots in the zebrafish are to the ones found in patients with this disease.

## Results

### The blood clots are formed in the CVP in *smarca5*$^{zko1049a}$

In our previously generated *smarca5*$^{zko1049a}$ mutants (*Ding et al., 2021*), we observed that the blood clots were formed in CVP at 2 days post fertilization (dpf), which was not present in their sibling embryos (*Figure 1A*). Our whole mount in situ hybridization (WISH) data showed that *scl* was expressed in blood clots, indicating that cells in the observed blood clots were primitive RBCs in *smarca5*$^{zko1049a}$ (*Figure 1B*). To directly observe the blood clot formation in the CVP, we used the transgenic line (Tg) (*gata1*:dsRed;*kdrl*:GFP) to label RBCs and endothelial cells, in *smarca5*$^{zko1049a}$ and in siblings. Confocal imaging analysis showed that the blood clots were formed inside the blood vessels (*Figure 1C*). Notably, there was no difference in the distribution of myeloid cells labeled by Tg (*coro1a*:GFP) or Tg (*mpo*:GFP) in caudal hematopoietic tissue (CHT) between *smarca5*$^{zko1049a}$ and their siblings, and we did not observe accumulation of myeloid cells in the blood clots of *smarca5*$^{zko1049a}$ (*Figure 1—figure supplement 1A*).

To further determine whether *smarca5* is involved in the development of primitive hematopoiesis, we examined the expression level of *gata1* and *pu.1*, which are the erythrocyte and myeloid marker genes, respectively, in *smarca5*$^{zko1049a}$ and their siblings. WISH and quantitative PCR (qPCR) analyses showed that the expression level of *gata1* and *pu.1* was comparable between *smarca5*$^{zko1049a}$ and their siblings at 33 hours post fertilization (hpf) (*Figure 1D–E* and *Figure 1—figure supplement 1B-C*) . Moreover, the expression level of *ikaros* and *scl*, which are two primitive erythrocyte markers, was normal (*Figure 1D–E*), as well as the expression of globin genes in *smarca5*$^{zko1049a}$ (*Figure 1E*). In addition, the myeloid markers *pu.1*, *lyz* and *mfap4* were normally expressed in *smarca5*$^{zko1049a}$ at 33 hpf and 2 dpf (*Figure 1—figure supplement 1B-C*). Thus, the early development of primitive erythrocytes and myeloid cells, is not affected upon the loss of *smarca5* in zebrafish embryos.

Taken together, these results show that *smarca5* is functionally required for normal behaviors of primitive erythrocytes and the blood clotting is formed by erythrocytes in *smarca5*$^{zko1049a}$.

### The blood clots are formed by RBC aggregation

To visualize how *smarca5*-deficient RBCs formed blood clots in the CVP of *smarca5*$^{zko1049a}$, we performed time lapse imaging using Tg (*gata1*:dsRed). We tracked the behavior of circulating RBCs in siblings (, *Video 1*) and *smarca5*$^{zko1049a}$ (*Video 2*) from 36 hpf to 2 dpf. The results showed that *smarca5*-deficient

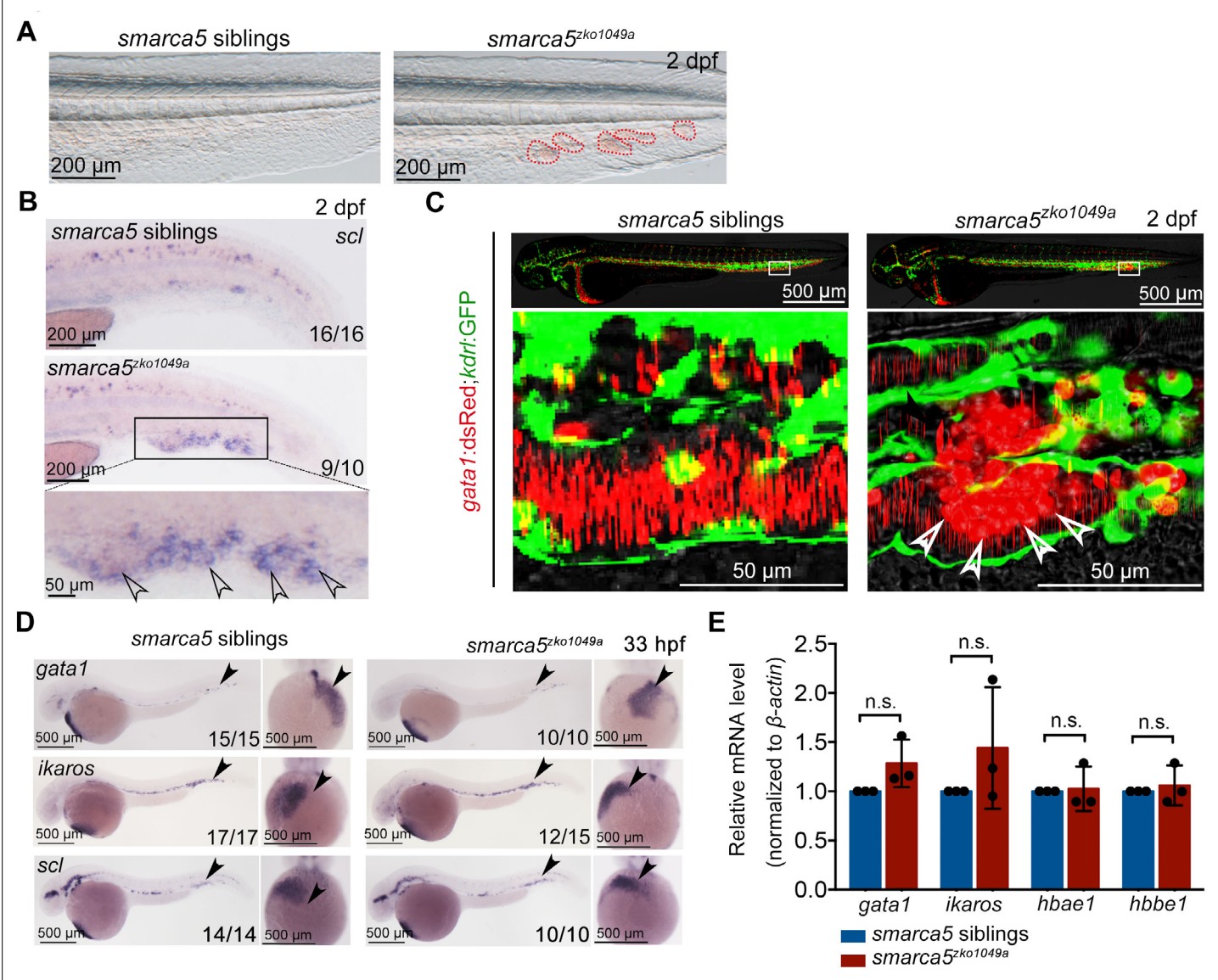

**Figure 1.** Blood clots occur in *smarca5*<sup>zko1049a</sup>. (**A**) The bright-field of tail region in *smarca5*<sup>zko1049a</sup> and their siblings at 2 days post fertilization (dpf). The areas circled by red dotted line show the blood clots in the caudal vein plexus. (**B**) Expression of *scl* at 2 dpf in *smarca5*<sup>zko1049a</sup> and their siblings by WISH. The amplification region in the black rectangular box in CHT shows the enriched expression of *scl* in blood clots (indicated by arrow heads) in the caudal vein plexus. (**C**) The confocal imaging of transgenic line (Tg) (*gata1*:dsRed;*kdrl*:GFP) in *smarca5*<sup>zko1049a</sup> and their siblings at 2 dpf. The amplification region in the white rectangular box in CHT shows the coagulation of red blood cells (RBCs) (indicated by arrow heads) in the caudal vein plexus. (**D**) Expression of *gata1*, *ikaros* and *scl* at 33 hr post fertilization (hpf) in *smarca5*<sup>zko1049a</sup> and their siblings by WISH. (**E**) qPCR analysis showing the expression of *gata1*, *ikaros*, *hbae1,* and *hbbe1* in *smarca5*<sup>zko1049a</sup> and their siblings at 33 hpf. The expression level of these genes in *smarca5* siblings was set at 1. Data are mean ± s.d. (**E**). Asterisk presents statistical significance (n.s. not significant). p Values were calculated by two-tailed unpaired Student's *t*-test.

The online version of this article includes the following figure supplement(s) for figure 1:

**Figure supplement 1.** *smarca5* is dispensable for the development of primitive myeloid cells.

---

RBCs tended to clump in the CVP at around 40 hpf, after which these clots will migrate or break off under blood flow at the early stage. As the blood clots formed with larger size, these clots will finally lodge in the vein (*Video 2*). The snapshot of Tg (*gata1*:dsRed) showed the process of blood clots formation from 36 hpf to 2 dpf in *smarca5*<sup>zko1049a</sup> and their siblings (*Figure 2A*). These results show that the clumping of RBCs precedes their sequestration in CVP, suggesting that the formation of blood clots might be independent of vascular niche.

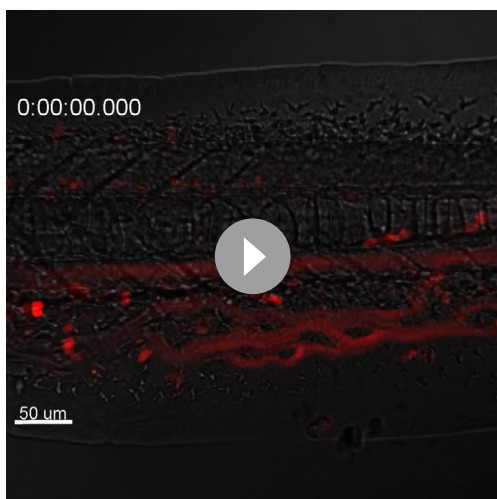

**Video 1.** The behavior of circulating RBCs in the tail region of sibling embryos. The time lapse imaging of sibling embryos with Tg (*gata1*:dsRed) background from 36 hpf to 2 dpf.

https://elifesciences.org/articles/72557/figures#video1

To further explore whether the blood clots formed in *smarca5*[zko1049a] were not resulted from the abnormal niche environment, we performed parabiosis experiment using *smarca5*[zko1049a] and their siblings. The parabiotic embryo pairs with *smarca5*[zko1049a] and siblings share a common bloodstream so that the behavior of circulating cells could reflect the influence of niche environment on these cells. We found that the blood clots, which occurred in *smarca5* mutants, were observed in both *smarca5*[zko1049a] and their siblings in parabiosis pairs (*Figure 2B*), indicating the *smarca5*-deficient RBCs form blood clots largely independent of niche environment. To specifically label the RBCs in *smarca5*[zko1049a] and their siblings, the Tg (*gata1*:dsRed) or Tg (*gata1*:GFP) transgenic line was used, respectively. The results showed that *smarca5*-deficient RBCs labeled by *gata1*:GFP aggregated both in *smarca5*[zko1049a] and in their siblings in parabiosis pairs (*Figure 2C*). Although several sibling RBCs labeled by *gata1*:dsRed were found trapped in blood clots, the vast majority of *gata1*:dsRed[+] cells were normally circulating in

blood stream both in *smarca5*[zko1049a] and their siblings (*Figure 2C*). Overall, these results indicate that the blood clots in *smarca5*[zko1049a] are formed largely in RBC-autonomous manner. To further explore whether thrombocytes participate in the formation of blood clots, we detected the blood clots using Tg (*CD41*:GFP). The imaging data showed that no *CD41*:GFP[high]-labeled thrombocytes were present in the blood clots (*Figure 2D*).

The CHT is a hematopoietic tissue critical for HSPC development. We thus wanted to know whether the blood clots formed in *smarca5*[zko1049a] could influence the structure of CHT, further leading to HSPC defects. As observed previously, the structure of CHT was normal in *smarca5*[zko1049a] and the number of *cmyb*:GFP[+] HSPCs in CHT at 2 dpf was comparable between *smarca5*[zko1049a] and their siblings (*Figure 2—figure supplement 1*) , indicating that the formation of blood clots in *smarca5*[zko1049a] is dispensable for HSPC development in CHT.

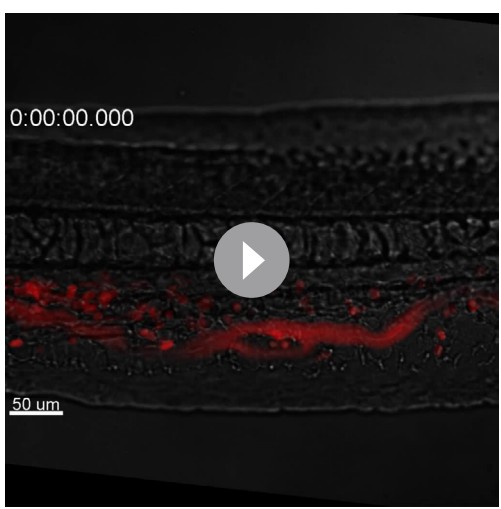

**Video 2.** The behavior of circulating RBCs in the tail region of *smarca5*[zko1049a]. The time lapse imaging of *smarca5*[zko1049a] with Tg (*gata1*:dsRed) background from 36 hpf to 2 dpf.

https://elifesciences.org/articles/72557/figures#video2

Zebrafish is a useful model to screen drugs for preclinical applications. In our *smarca5*-deficient zebrafish model, we observed blood clots in veins, raising questions regarding whether there was a thrombus-like phenotype. To this end, we tried to test the clinically used anti-thrombosis drugs to treat *smarca5*[zko1049a] embryos. We tested reagents including heparin, aspirin, and argatroban that have been reported to target thrombosis to examine whether the blood clots in *smarca5*[zko1049a] can be alleviated after chemical treatment. The embryos were incubated in aspirin or injected with heparin or argatroban at 36 hpf and the phenotype was examined at 2 dpf. As a result, we found that a direct thrombin inhibitor, argatroban, but not an antithrombin-dependent drug, heparin, or a platelet aggregation inhibitor, aspirin, partially prevented blood clot formation in *smarca5*[zko1049a] at 2 dpf (*Figure 2E-G*). These results suggest that the RBC aggregation in *smarca5*[zko1049a] is more relevant to venous thrombosis

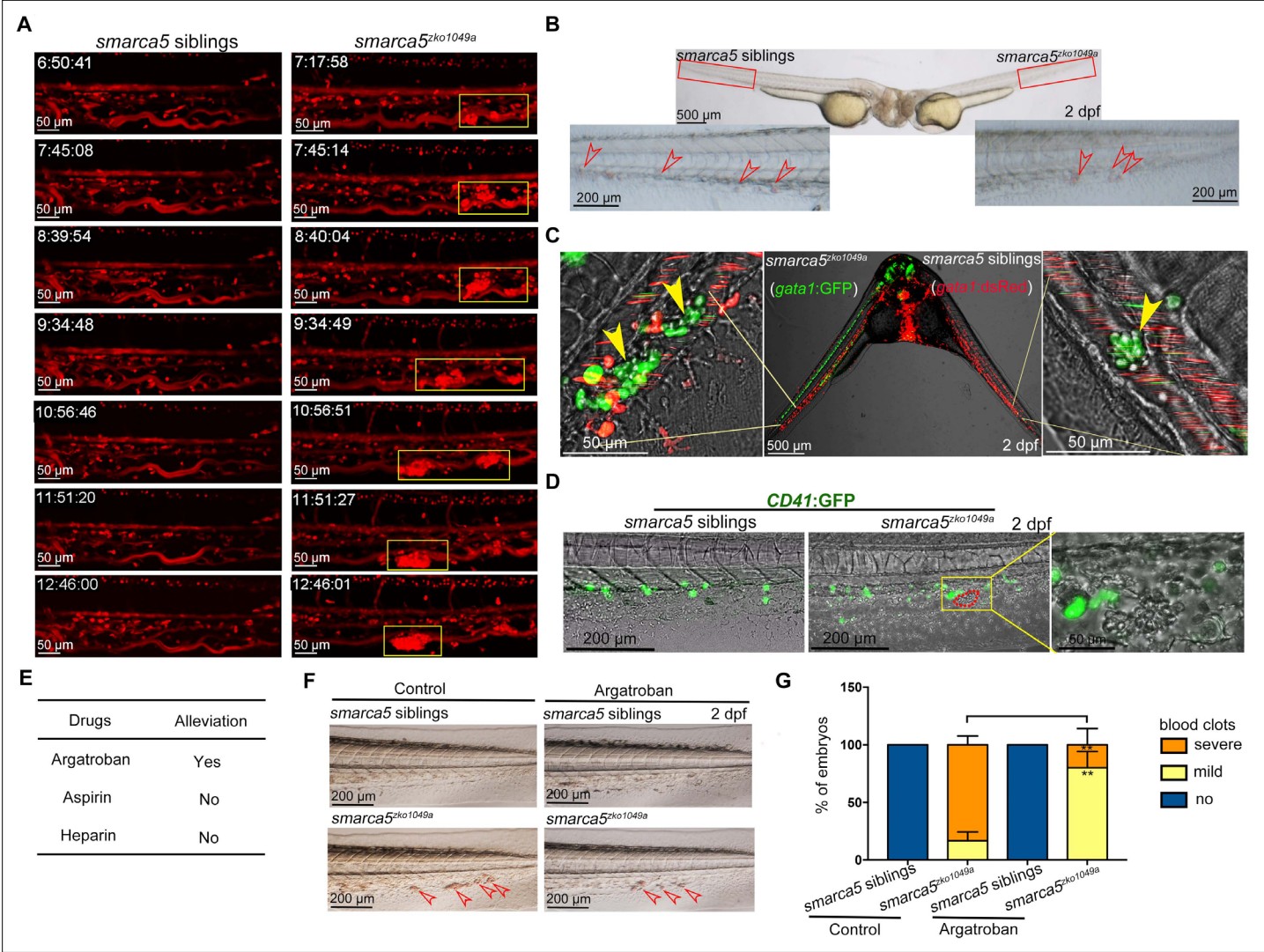

**Figure 2.** The blood clots are formed by RBC aggregation. (**A**) The snapshot of Tg (*gata1*:dsRed) in *smarca5^zko1049a^* and their siblings from 36 hpf to 2 dpf. The yellow rectangular boxes in CHT show the formed blood clots *smarca5^zko1049a^*. (**B**) The bright-field of parabiosis at 2 dpf between *smarca5^zko1049a^* and their siblings. The amplification region in the red rectangular box in CHT shows the blood clots (indicated by arrow heads) in *smarca5^zko1049a^* and their siblings. (**C**) The confocal imaging of parabiosis generated between *smarca5^zko1049a^* and their siblings with Tg (*gata1*:GFP) and Tg (*gata1*:dsRed) background, respectively. The magnification in CHT shows the aggregation of *gata1*:GFP⁺ cells (indicated by arrow heads) in the caudal vein plexus. (**D**) The confocal imaging of Tg (*CD41*:GFP) in *smarca5^zko1049a^* and their siblings at 2 dpf. The magnification in the yellow rectangular box in CHT shows the blood clots and the distribution of *CD41*:GFP⁺ cells. (**E**) Drugs used to examine whether the blood clots in *smarca5^zko1049a^* can be alleviated. (**F**) The bright-field of tail region in *smarca5^zko1049a^* at 2 dpf in control group and with argatroban treatment. The blood clots are indicated by arrow heads. (**G**) The quantification of blood clots phenotype in (**F**). Data are mean ± s.d. (**G**). Asterisk presents statistical significance (**p< 0.01). p Values were calculated by two-tailed unpaired Student's *t*-test.

The online version of this article includes the following figure supplement(s) for figure 2:

**Figure supplement 1.** The RBC aggregation has no overt influence on the number of HSPCs in the CHT.

and the *smarca5*-deficient zebrafish model may serve as a venous thrombosis model to screen drugs in preclinical setting.

## *Smarca5*-deficient RBCs manifest disintegration of cristae in mitochondria

Both quantitative and qualitative changes in RBCs have been linked to thrombosis (**Weisel and Litvinov, 2019**). To identify whether *smarca5* deletion will lead to the quantitative changes of RBCs, we performed fluorescence activating cell sorter (FACS) analysis of the percentage of *gata1*:dsRed⁺

cells in *smarca5*[zko1049a] and their siblings. Deletion of *smarca5* did not lead to the significant changes in RBC counts at 2 dpf (*Figure 3—figure supplement 1A-B*). These data suggest that the blood clots in *smarca5*[zko1049a] are formed by RBC aggregation with no overt cell number change.

To explore whether there exist qualitative changes in *smarca5*-deficient RBCs, we performed blood-smear and Giemsa-staining analysis. The results showed that the morphology of RBCs had no obvious changes in *smarca5*[zko1049a] (*Figure 3—figure supplement 1C*). And the statistical analysis showed that the nucleocytoplasmic ratio was normal in *smarca5*-deficient RBCs (*Figure 3—figure supplement 1D*), indicating that the differentiation of RBCs at 2 dpf was not evidently impaired upon *smarca5* loss.

To further investigate the changes in subcellular structure of erythrocytes in *smarca5*[zko1049a], we performed TEM analysis. Compared with *smarca5* sibling embryos in which the circulating RBCs had normal organization in mitochondria (*Figure 3A–B*), we found that the *smarca5*-deficient erythrocytes displayed disintegration of cristae in mitochondria while nuclear integrity was preserved in *smarca-5*[zko1049a] (*Figure 3C–E*). The area of mitochondria was not significantly changed and the number of mitochondria was slightly increased but not significantly changed in *smarca5*-deficent RBCs (*Figure 3F–G*). We propose that the erythrocytes in *smarca5*[zko1049a] may have undergone cellular damages, such as oxidative stress, which could lead to the disintegration of mitochondria (*Lewerenz et al., 2018*). It is also possible that the mitochondrial defects may further exacerbate oxidative stresses (*Dan Dunn et al., 2015*; *Yang et al., 2016*), thereafter leading to the erythroid defects caused by *smarca5* deletion. Thus, the morphological disruption in mitochondria suggests the disorder of cellular homeostasis in erythrocytes after *smarca5* deletion.

## Transcriptional dysregulation of genes related to erythrocyte function and homeostasis after *Smarca5* deletion

Smarca5 typically regulates nucleosome spacing, further affecting gene transcription (*Clapier et al., 2017*). To decipher how loss of Smarca5 affects the transcriptome, RNA-seq was used to profile sorted erythrocytes labeled by *gata1*:dsRed from *smarca5*[zko1049a] and their siblings at 2 dpf, respectively (*Figure 4A*). Principal components analysis (PCA) indicated clear separation of the *smarca5*[zko1049a] and sibling samples (*Figure 4—figure supplement 1A*). A total of 1506 genes were upregulated and 633 genes were downregulated significantly (Log2(fold change) > 1, adjusted p-value < 0.05) in *smarca5*-deficient erythrocytes compared to erythrocytes from siblings (*Figure 4B*).

Gene set variation analysis (GSVA) revealed a strong enrichment of terms related to 'Gata1 targets', 'autophagy', 'erythrocytes take up carbon dioxide and release oxygen' and 'erythrocytes take up oxygen and release carbon dioxide' in sibling erythrocytes; for *smarca5*[zko1049a], while the 'apoptosis', 'environmental stress response', 'senescence', and 'cell oxidation' were markedly increased (*Figure 4C*). The enrichment plots showed the decreased expression of genes related to 'erythrocyte homeostasis' in *smarca5*[zko1049a], whereas the expression of genes related to 'inflammatory response' was increased (*Figure 4D*). These results suggest that the disrupted pathways in *smarca5*-deficient RBCs were highly related to erythrocyte function and cellular homeostasis.

RBCs have specialized proteome, which is enriched in hemoglobin. We then focused on the expression of hemoglobin complex related genes. The expression level of embryonic globin genes, including *hbae1*, *hbae3*, *hbbe1*, *hbbe2*, and *hbbe3*, was not obviously affected in *smarca5*[zko1049a] and the expression level of the adult globin genes, including *hbaa1*, *hbba1*, and *hbba2*, was comparable between *smarca5*[zko1049a] and their siblings (*Figure 4—figure supplement 1B*). The WISH results also showed the comparable expression of embryonic and adult globin genes in *smarca5*[zko1049a] and their siblings (*Figure 4—figure supplement 1C*). Moreover, the level of hemoglobin detected by O-dianisidine staining was comparable between *smarca5*[zko1049a] and their siblings (*Figure 4—figure supplement 1D*). Therefore, *smarca5* deletion does not lead to obvious hemoglobinopathy in *smarca5*[zko1049a] at 2 dpf and the *smarca5*-deficient RBCs does not have the obvious developmental delay.

In addition, we observed the persistent expression of *spi1a*, *spi1b*, *mfap4*, and *lyz* markers characteristic of myeloid cells in *smarca5*-defecient erythrocytes (*Figure 4—figure supplement 1E*). Perturbation of the exquisite control by *smarca5* likely causes 'hybrid' primitive erythrocytes that resemble partial transcriptional properties of myeloid cells. One possible mechanism for this phenotype is the regulation of SMARCA5 and CTCF at the enhancer of PU.1 (*Dluhosova et al., 2014*), thereby blocking of *smarca5* leads to the upregulation of *pu.1* gene expression. However, despite the inappropriate expression of myeloid genes in *smarca5*-deficient RBCs, the development of myeloid lineage was not

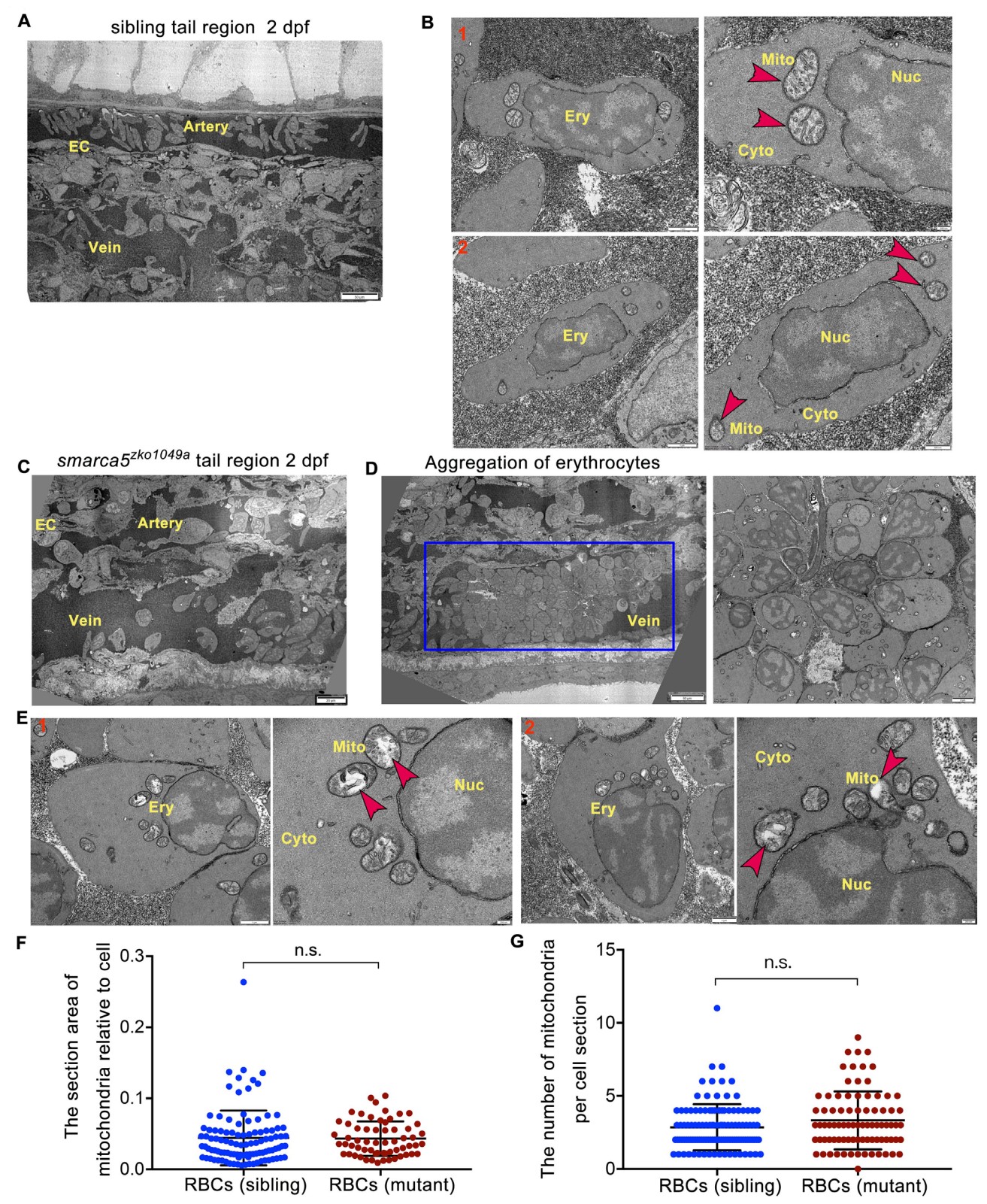

**Figure 3.** TEM shows abnormal organelle morphology in *smarca5*-deficient RBCs. (**A**) The transmission electron microscopy (TEM) view of a longitudinal section through the artery and vein plexus in sibling tail region at 2 dpf. (**B**) TEM view of erythrocytes in sibling embryos. The red arrow heads indicate the characteristic pattern of organization in mitochondria. (**C**) TEM view of a longitudinal section through the artery and vein plexus in *smarca5^zko1049a^* tail region at 2 dpf. (**D**) The blue rectangular box showing blood clots in the caudal vein plexus in *smarca5^zko1049a^*. The magnification of blood clots is

*Figure 3 continued on next page*

*Figure 3 continued*

shown (right). (**E**) TEM view of erythrocytes in smarca5$^{zko1049a}$. The red arrow heads indicate the characteristic pattern of organization in mitochondria. The disintegration of cristae in mitochondria is observed in smarca5-deficient RBCs. Ery, erythrocyte; EC, endothelial cell; Mito, mitochondria; Nuc, nucleus; Cyto, cytoplasm. (**F**) Scatter plots showing the section area of mitochondria relative to cell in RBCs from smarca5$^{zko1049a}$ and their siblings. (**G**) Scatter plots showing the number of mitochondria per cell section in RBCs from smarca5$^{zko1049a}$ and their siblings. Data are mean ± s.d. (**F and G**). Asterisk presents statistical significance (n.s. not significant). p Values were calculated by two-tailed unpaired Student's t-test.

The online version of this article includes the following figure supplement(s) for figure 3:

**Figure supplement 1.** The morphology and number of RBCs has no obvious change in smarca5$^{zko1049a}$.

obviously impaired in smarca5$^{zko1049a}$ manifested with normal expression pattern of pu.1 and lyz at 33 hpf and 2 dpf (*Figure 1—figure supplement 1B-C*), suggesting the unaltered lineage choices at the primitive stage. To further explore whether the inappropriate expression of myeloid genes in smarca5-deficient RBCs caused RBC aggregation, we tried to knockdown of pu.1 in smarca5$^{zko1049a}$. The results showed that knockdown of pu.1 cannot rescue the RBC aggregation phenotype in smarca5$^{zko1049a}$ (*Figure 4—figure supplement 1F-G*).

Taken together, smarca5 deletion leads to the disrupted pathways related to erythrocyte function and cellular homeostasis.

## Deletion of *Smarca5* disrupts chromatin accessibility in RBCs

To explore the mechanism through which Smarca5 in regulating the chromatin accessibility in RBCs, we performed the ATAC-seq in FACS-purified RBCs from smarca5$^{zko1049a}$ and their siblings at 2 dpf. Density heatmaps of mapped ATAC-seq reads showed that fragments less than 100 bp in length clustered immediately upstream of transcriptional start sites (TSSs) throughout the zebrafish genome in both mutant and sibling RBC nuclei (*Figure 5—figure supplement 1A-B*). The PCA analysis was performed for ATAC-seq samples and the results showed that the mutant samples or sibling samples can be grouped together, respectively (*Figure 5—figure supplement 1C*). The feature distributions of mutant-ATAC-seq peaks and sibling-ATAC-seq peaks across the genome were identified by ChIP-seeker (*Figure 5—figure supplement 1D*).

We then calculated the number of genes with changes in chromatin accessibility after smarca5 deletion (*Figure 5A*). The chromatin accessibility at promoters of 256 genes was decreased in smarca5$^{zko1049a}$, while there were 439 genes with increased chromatin accessibility at promoters after smarca5 deletion. Next, we screened the motifs enriched in sibling RBC-specific accessible chromatin regions. We found that the erythrocyte master regulator-Gata1 motif was on the top list (*Figure 5B*). Thus, deletion of smarca5 might affect the binding of hematopoietic transcription factors in erythrocytes, such as Gata1. It has been reported that Smarca5 could interact with Gata1 in erythrocytes (*Rodriguez et al., 2005*). We propose that Smarca5 might be recruited by Gata1 and mediate the chromatin accessibility of Gata1-binding sites in target genes.

We further detected the genes in which the chromatin accessibility at promoters or distal regions and their transcription were both increased or decreased after smarca5 deletion (*Figure 5C* and *Figure 5—figure supplement 1E*). The results showed that the chromatin accessibility at promoters and transcription of 84 genes, such as il34, cox4i2, skap2, vclb, and acbd7, were increased, while the chromatin accessibility at promoters and transcription of 36 genes, such as trim2a, keap1a, acox3, igfbp1a, and ada, were decreased in smarca5-deficient RBCs (*Figure 5D*). The lack of overlap between changes in gene expression and ATAC-seq signals may partially due to the complex interactions between cis-regulatory elements and trans-regulatory elements in the regulation of gene expression (*Gibson and Weir, 2005*; *Hill et al., 2021*; *Wittkopp, 2005*; *Wittkopp et al., 2004*). Moreover, cells exhibit significant variations in gene expression and the underlying regulation of chromatin because of intrinsic and extrinsic factors (*Ma et al., 2020*). The accessibility of peaks and the expression of genes are not exactly matched, which may contribute to explaining the lack of overlap between changes in gene expression and ATAC-seq signals. Taken together, smarca5 deletion leads to the disrupted chromatin accessibility and transcriptome in RBCs.

## *Keap1a* acts as a downstream target of Smarca5 in RBC aggregation

Based on the screening results, the chromatin accessibility at keap1a promoters, which contains Gata1 motif, was decreased in smarca5$^{zko1049a}$ (*Figure 6A*). The transcription level of keap1a detected by

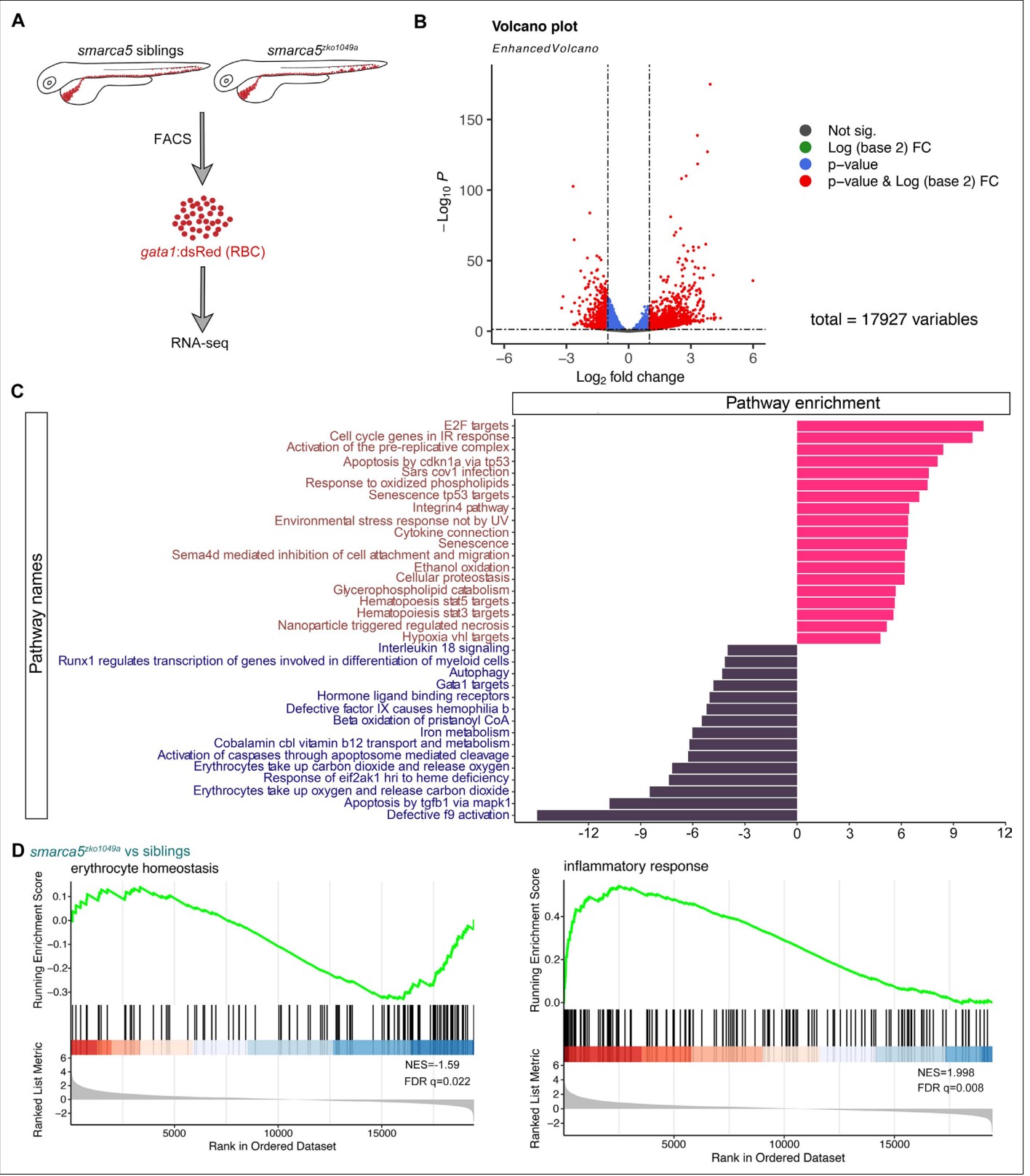

**Figure 4.** Transcriptional disruption of genes related to erythrocyte function and homeostasis after *smarca5* deletion. (**A**) Schematic representation of the RBC RNA-seq workflow in *smarca5^zko1049a^* and their siblings at 2 dpf. (**B**) Volcano plot showing differential expression genes between RBCs from *smarca5^zko1049a^* and their siblings by DESeq2. −Log10 P, negative log10 adjusted p-value. Adjusted p-value < 0.05, log2 fold change >1. (**C**) Differential pathways enriched in RBCs from *smarca5^zko1049a^* and their siblings by GSVA. The x axis represents the t values of GSVA scores. Two-sided unpaired limma-

*Figure 4 continued on next page*

*Figure 4 continued*

moderated t test. (**D**) Enrichment plots for the top pathways in the mutant RBCs by GSEA (Gene Set Enrichment Analysis).

The online version of this article includes the following figure supplement(s) for figure 4:

**Figure supplement 1.** RNA-seq analysis for RBCs in *smarca5*^*zko1049a*^ and their siblings.

qPCR was also decreased in *smarca5*-deficient RBCs (*Figure 6B*). Given that *keap1* was previously identified to correlate with human venous thrombosis (*Akin-Bali et al., 2020*), we propose that *keap1a* may act as a downstream target of Smarca5 in RBCs.

Keap1-Nrf2 system is an evolutionarily conserved defense mechanism in oxidative stress (*Itoh et al., 1997*; *Itoh et al., 1999*). In cytoplasm, Keap1 could anchor to Nrf2 to facilitate the Nrf2 degradation, while oxidative stress leads to the proteasomal degradation of Keap1 and release of Nrf2 to

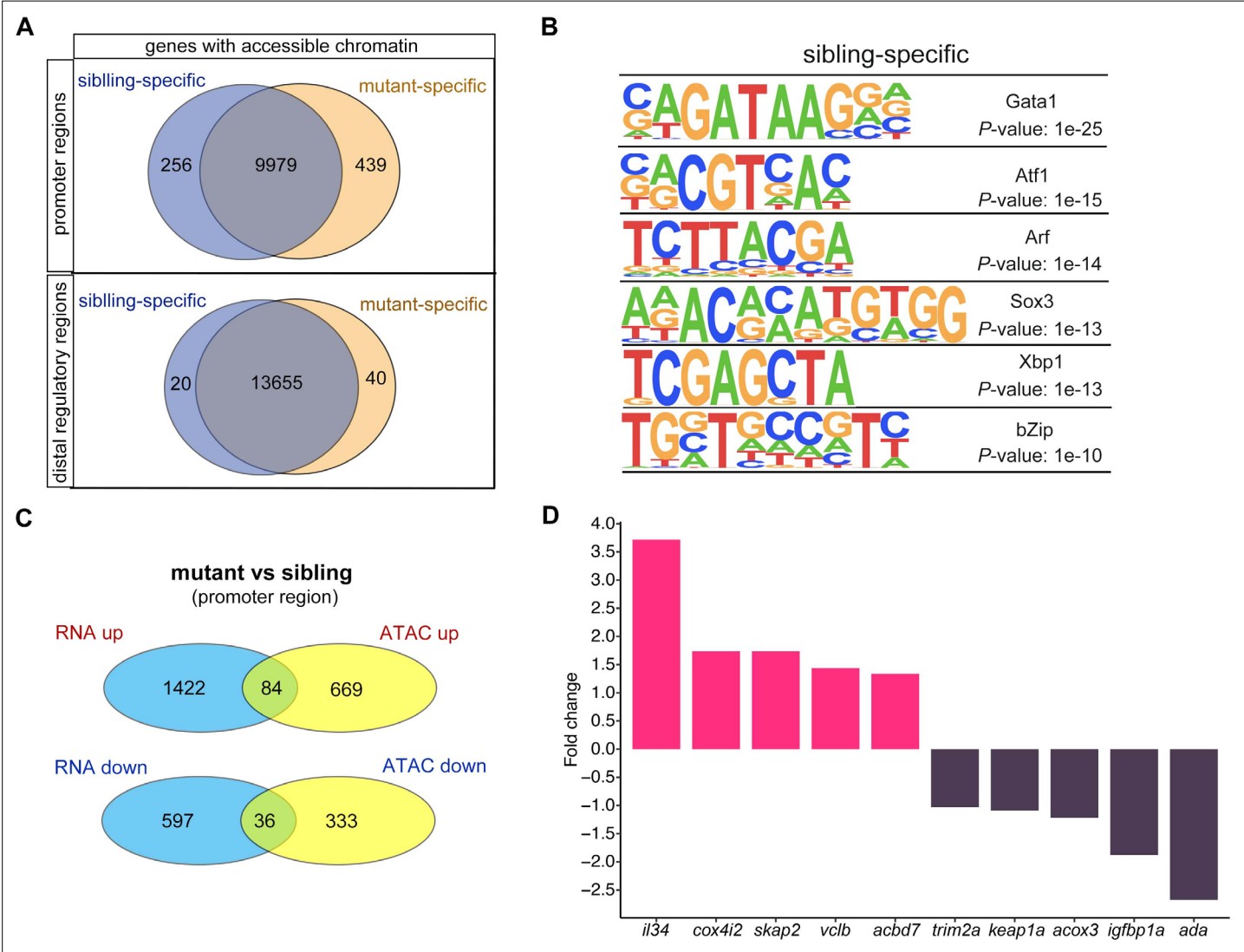

**Figure 5.** Changes in chromatin accessibility in RBCs from *smarca5*^*zko1049a*^ and their siblings. (**A**) Venn plot showing the genes of sibling and mutant specific accessible chromatin regions at promoter and distal regulator regions, respectively. (**B**) Motifs enriched in nucleosome-free regions (NFRs) with lost accessibility after smarca5 deletion. (**C**) Venn plot showing the overlap of genes with specific accessible chromatin at promoter regions and upregulated expression in mutant (top) and sibling (bottom) RBCs, respectively. Genes for ATAC-seq were assigned by differential accessible regions. (**D**) Bar plot showing the selected sibling and mutant-specific differential expression genes. Fold change, $\log_2$ fold change.

The online version of this article includes the following figure supplement(s) for figure 5:

**Figure supplement 1.** ATAC-seq analysis for RBCs in *smarca5*^*zko1049a*^ and their siblings.

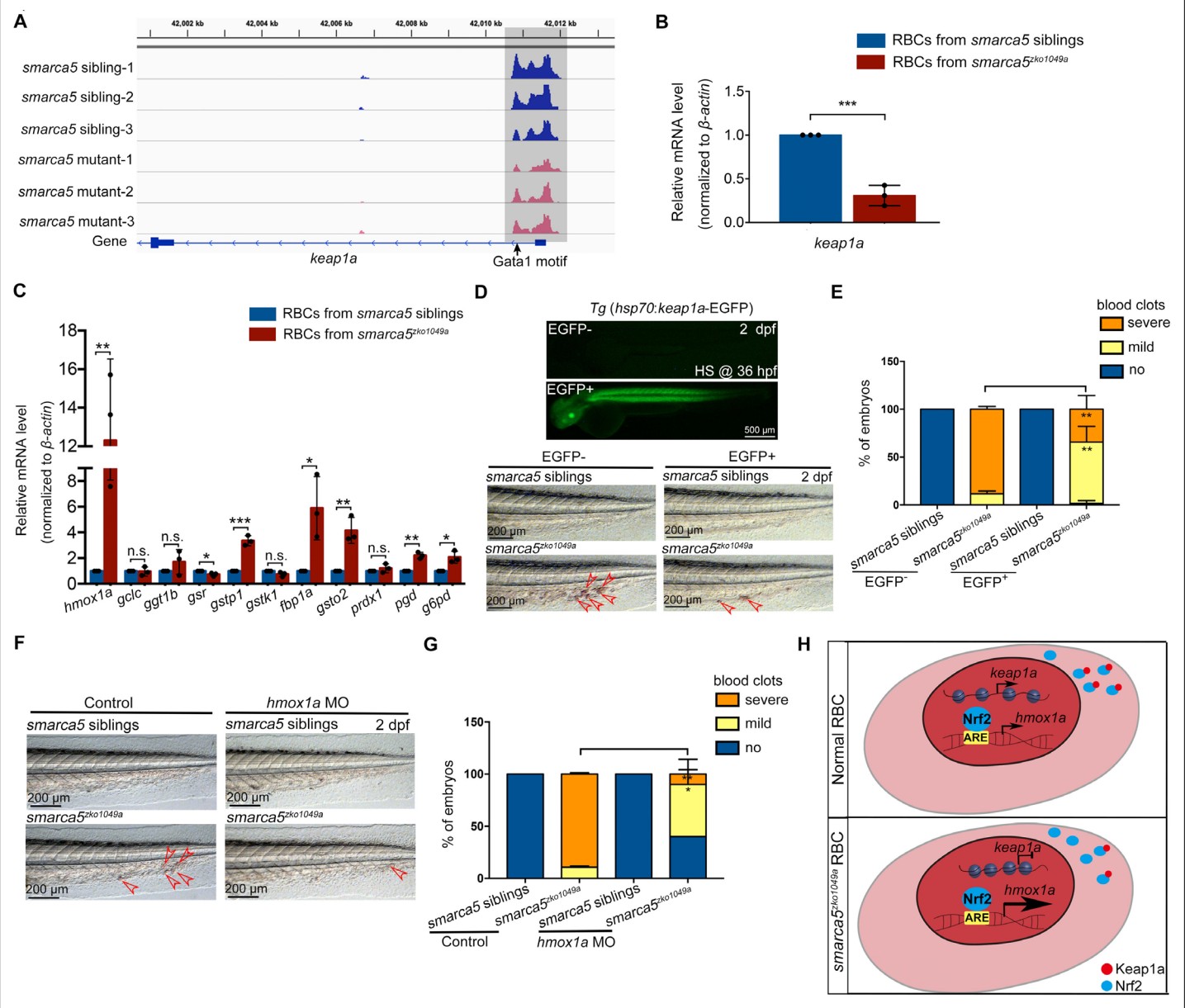

**Figure 6.** Keap1-Nrf2 signaling pathway acts at downstream of Smarca5 in regulating RBC aggregation. (**A**) The browser views showing the ATAC-seq peaks in *keap1a* promoter with in *smarca5^zko1049a* and their siblings. Gray box indicates the change of ATAC-seq peaks after *smarca5* deletion. The location of Gata1 motif at *keap1a* promoter is indicated by arrow. (**B**) qPCR analysis showing the expression of *keap1a* in RBCs from *smarca5^zko1049a* and their siblings at 2 dpf. (**C**) qPCR analysis showing the expression of *hmox1a*, *gclc*, *ggt1b*, *gsr*, *gstp1*, *gstk1*, *fbp1a*, *gsto2*, *prdx1*, *pgd* and *g6pd* in RBCs from *smarca5^zko1049a* and their siblings at 2 dpf. (**D**) The imaging of EGFP fluorescence in Tg (*hsp70:keap1a*-EGFP) embryos at 2 dpf. Heat shock was performed at 36 hpf. The bright-field of tail region in *smarca5^zko1049a* and their siblings, with or without Smarca5 overexpression at 2 dpf. (**E**) The quantification of blood clots phenotype in (**D**). (**F**) The bright-field of tail region in *smarca5^zko1049a* and their siblings, in control group and with *hmox1a* MO injection. The blood clots are indicated by arrow heads. (**G**) The quantification of blood clots phenotype in (**F**). (**H**) Schematic representation of Smarca5 in regulating erythrocyte aggregation via Keap1-Nrf2 signaling. In cytoplasm, Keap1 could anchor to Nrf2 to facilitate the Nrf2 degradation, while the release of Nrf2 to the nucleus could activate the expression of oxidation defense factors. In *smarca5*-deficient RBCs, the chromatin accessibility at *keap1a* promoters and the transcription of *keap1a* were decreased, which led to the excessive activation of *hmox1a*. Data are mean ± s.d. (**B, C, E, G**). Asterisk presents statistical significance (*p < 0.05, **p < 0.01, ***p < 0.001, n.s. not significant). p Values were calculated by two-tailed unpaired Student's *t*-test.

The online version of this article includes the following source data and figure supplement(s) for figure 6:

**Figure supplement 1.** The upregulation of *HMOX1* in hemin-induced K562 cells after knockdown of *SMARCA5*.

**Figure supplement 1—source data 1.** Full raw unedited blots from *Figure 6—figure supplement 1D*.

*Figure 6 continued on next page*

*Figure 6 continued*

**Figure supplement 1—source data 2.** Uncropped blots from *Figure 6—figure supplement 1D* with the relevant bands clearly labelled.

**Figure supplement 2.** The free radical generation may play a major role in RBC aggregation in *smarca5^zko1049a^*.

the nucleus, thereafter activate the expression of oxidation defense factors. Both our RNA-seq and qPCR analysis showed the downregulation of *keap1a* and as a downstream target of Nrf2, *hmox1a* showed a markedly increase in gene expression upon *smarca5* deletion (*Figure 6C*), suggesting the disruption of Keap1-Nrf2 signaling pathway. It is worthy of note that, although the upregulated expression of Keap1-Nrf2 downstream targets can protect cells from oxidative damage, the excessive activation of *hmox1a*, which catalyzes the degradation of heme to biliverdin, carbon monoxide, and Fe2+, could even lead to the oxidative stress (*Hassannia et al., 2019*). Thus, we propose that the unbalanced Keap1-Nrf2 signaling, especially the upregulation of *hmox1a*, could increase oxidative damage in *smarca5*-deficient RBCs. We next performed functional validation of *keap1a* in *smarca5^zko1049a^* by overexpression of *hsp70:keap1a*-EGFP. Heat shock was performed at 24 hpf and 36 hpf, and the phenotype was examined at 2 dpf. The results showed that overexpression of *keap1a* in *smarca5^zko1049a^* could partially rescue the blood clots phenotype (*Figure 6D–E*). In addition, knockdown of *hmox1a*, the downstream target of Keap1-Nrf2, can also partially rescue the blood clots phenotype in *smarca5^zko1049a^* (*Figure 6F–G*), further supporting that the Keap1-Nrf2 signaling pathway downstream of Smarca5 is essential for blood clot formation (*Figure 6H*).

To further identify the conserved role of *SMARCA5* in mammalian erythrocyte homeostasis, we used K562 cells (human erythroleukemic cells) to perform further analysis. Treatment of hemin induced the hemoglobinization of most K562 cells, suggesting the efficient erythroid differentiation (*Figure 6—figure supplement 1*). We then knocked down *SMARCA5* in hemin-induced K562 cells using *SMARCA5* short interfering RNA (siRNA) and the qPCR and western blot analyses showed that both the RNA and protein levels of *SMARCA5* were decreased significantly after si*SMARCA5*s (si*SMARCA5*-1, si*SMARCA5*-2 and si*SMARCA5*-3) transfection (*Figure 6—figure supplement 1*). In addition, the expression of *KEAP1* was decreased while *HMOX1* was obviously upregulated after *SMARCA5* knockdown (*Figure 6—figure supplement 1*), indicating the conserved role of *SMARCA5* in human erythrocyte homeostasis.

Considering the role of Keap1-Nrf2 signaling pathway in oxidative stress regulation, we further asked whether the oxidative stress could be a trigger for blood clot formation in *smarca5* mutants. Then, we used a free radical scavenger glutathione to determine the mechanisms of *smarca5*-deficiency induced blood clots. We found that glutathione obviously prevented RBC aggregation in *smarca5^zko1049a^* (*Figure 6—figure supplement 2*), implying that free radical generation may play an important role in RBC aggregation in *smarca5^zko1049a^*.

Taken together, loss of *smarca5* leads to the disruption of *keap1a* expression and excessive activation of *hmox1a* in *smarca5^zko1049a^*, which together contribute to the formation of blood clots.

## Discussion

In this work, we develop a zebrafish RBC aggregation model with a deletion of an epigenetic regulator-*smarca5*. The blood clots are formed in the CVP of *smarca5^zko1049a^* and the erythrocytes manifest disintegration of cristae in mitochondria. Further transcriptome and chromatin accessibility analysis show that *keap1a* acts as a downstream target of Smarca5. Moreover, the elevated expression of the downstream target of Keap1-Nrf2, *hmox1a*, leads to the aggregation of *smarca5*-deficient RBCs. Together, these results demonstrate the protective role of Smarca5 in regulating erythrocyte homeostasis and that the *smarca5* loss-of-function zebrafish mutant may serve as a new thrombosis model to screen molecular drugs for clinical therapy.

Considering the conserved coagulation and anticoagulation signaling pathway, the zebrafish model has been used to study the physiology of thrombosis (*Hanumanthaiah et al., 2002*; *Jagadeeswaran et al., 1999*; *Sheehan et al., 2001*). The ferric chloride and laser injury methods are widely used in zebrafish to generate thrombus in the circulation (*Gregory et al., 2002*). Phenylhydrazine-treated zebrafish also develop thrombosis in the caudal vein (*Zhu et al., 2016*). Moreover, zebrafish is an ideal model to explore novel players in thrombosis based on genetic manipulation. For example, mutation of *anti-thrombin III* gene in zebrafish can mimic disseminate intravascular coagulation (*Liu et al.,*

2014). miR-126 was identified as a regulator of thrombi generation in zebrafish (*Zapilko et al., 2020*). Importantly, the transparency of zebrafish embryo makes it feasible to image the kinetics of thrombus formation. In our study, the *gata1*:dsRed-labeled RBCs were imaged during blood clot formation. Thus, the zebrafish thrombosis model is a great asset for exploring the underlying mechanisms in thrombosis formation.

Unlike Brg1, which is essential for mouse erythrocyte development by regulating globin gene expression (*Bultman et al., 2005*; *Griffin et al., 2008*), Smarca5 is required for primitive erythrocyte homeostasis at the erythrocyte differentiation stage. Deletion of *smarca5* does not lead to the gross changes in RBC morphology and viability, but specifically results in the RBC aggregation phenotype. The mechanistic details for different chromatin remodelers functioning in the different processes during erythropoiesis warrant further investigation.

Previous evidence suggests that chromatin remodeler NuRD is required to maintain lineage fidelity during erythroid-megakaryocyte ontogeny (*Gao et al., 2010*; *Gregory et al., 2010*). Our results show that, despite the normal lineage choice for primitive erythrocytes in *smarca5$^{zko1049a}$*, the aberrant activation of myeloid genes occurred in RBCs after *smarca5* deletion. The exquisite cell lineage control by *smarca5* may be due to the regulation of SMARCA5 at the enhancer of PU.1 (*Dluhosova et al., 2014*).

The RBCs are sensitive to mitochondrial biogenesis and function. A previous study shows that during human erythrocyte specification, the pathways related to mitochondrial biogenesis are enhanced through post-transcriptional regulation (*Liu et al., 2017*). Deletion of mitochondria factors resulted in metabolic changes and histone hyperacetylation, further leading to the impaired erythrocyte differentiation. Moreover, the transcription elongation factor TIF1γ directly regulates mitochondrial genes and histone methylation during erythrocyte differentiation (*Rossmann et al., 2021*). These studies suggest that the mitochondria biogenesis and function are highly regulated during normal erythropoiesis through transcriptional, epigenetic and post-transcriptional mechanisms, which may explain the specific defects observed in *smarca5*-deficient erythroid cells.

Besides the conserved role of Keap1-Nrf2 system in oxidative stress, Keap1-Nrf2 is also demonstrated to act as a regulator in cell development and differentiation across multiple tissues and cell types. For instance, Keap1-Nrf2 signaling pathway is indispensable for hematopoietic stem cell (HSC) lineage commitment in mice (*Murakami et al., 2014*). Knockout of *Keap1* in HSCs showed enhanced granulocyte-monocyte differentiation ability at the expense of lymphoid and erythrocyte differentiation. And the expression level of erythrocyte and lymphoid genes was decreased in *Keap1*-deficient HSCs. Importantly, the abundance of Hmox1 is upregulated during erythrocyte differentiation, and Hmox1 expression must be tightly regulated at appropriate level for efficient erythropoiesis (*Garcia-Santos et al., 2014*). Overexpression of Hmox1 impairs hemoglobin synthesis, while lack of Hmox1 leads to the enhancement of hemoglobinization. Here, we show that the disruption of *keap1a* expression and excessive activation of *hmox1a* in *smarca5$^{zko1049a}$* contribute to RBC aggregation. Besides the free radical generation, which may play an important role in RBC aggregation in *smarca5$^{zko1049a}$*, we cannot rule out other possibilities that may also be involved in the observed phenotype, such as the regulation of Keap1-Nrf2 signaling pathway in erythrocyte gene expression.

In summary, we have demonstrated, for the first time, that deletion of *smarca5* in zebrafish leads to the RBC aggregation by regulating the Keap1-Nrf2 signaling pathway in RBCs. These findings raise the possibility that zebrafish *smarca5* mutant may serve as a new venous thrombosis model for drug screening and pre-clinical therapeutic assessment.

## Materials and methods
### Zebrafish strains
Zebrafish strains including Tubingen, Tg (*CD41*:GFP) (*Lin et al., 2005*), Tg (*gata1*:dsRed) (*Traver et al., 2003*), Tg (*kdrl*:mCherry) (*Bertrand et al., 2010*), Tg (*gata1*:dsRed;*kdrl*:GFP) (kindly provided by Stefan Schulte-Merker (Hubrecht Institute, Utrecht, The Netherlands)), Tg (*mpo*:GFP) (*Renshaw et al., 2006*), Tg (*coro1a*:GFP) (*Li et al., 2012*), *smarca5$^{zko1049a}$* heterozygous mutants (*Ding et al., 2021*) were raised under standard conditions (28.5 °C in system water). The zebrafish embryos were raised in incubator at 28.5 °C. The present study was approved by the Ethical Review Committee of the Institute of Zoology, Chinese Academy of Sciences, China.

## Cell line

The human K562 cells (ATCC:CCL-243) were cultured in RPMI-1640 medium supplemented with 10 % FBS at 37 °C in 5 % $CO_2$.

## Whole mount in situ hybridization (WISH)

WISH was performed as previously described (*Wang et al., 2011*). The Digoxigenin-labeled RNA probe genes including *gata1, ikaros, scl, pu.1, lyz, hbae1, hbae3, hbbe1, hbbe2, hbbe3, hbaa1, hbba1,* and *hbba2* were cloned from zebrafish cDNA and ligated to the T-vector, then in vitro transcribed using T7 or SP6 polymerase.

## Quantitative PCR (QPCR)

Total RNAs were extracted from *smarca5$^{zko1049a}$* and their sibling embryos using TRIzol reagent (Life technologies, 15596018) or from sorted RBCs using QIAGEN RNeasy Mini Kit (Cat. No. 74104). The cDNA was reverse transcribed using M-MLV Reverse Transcriptase (Promega, M1701). The detailed primers used for qPCR are listed in *Supplementary file 1A*.

## Morpholinos (MOs)

The antisense MOs were purchased from GeneTools. The sequences of MOs were used as previous described, these gene-specific MOs include *hmox1a* MO and *pu.1* MO. The detailed sequence and dosage used in this work are listed in *Supplementary file 1B*.

## Parabiosis experiment

Parabiosis experiment was performed by following the previous published procedures (*Demy et al., 2013*; *Hagedorn et al., 2016*). Briefly, *smarca5$^{zko1049a}$* and their sibling embryos between the 128 cell blastula and 30 % epiboly stages were removed out of chorions and gently transferred into methylcellulose drop under fish water. Then, detach a few cells from each embryo at the contact points using the pulled glass micropipette and move these two embryos contact each other properly until they fusion together.

O-dianisidine staining, Giemsa-staining, and Benzidine staining *smarca5$^{zko1049a}$* and their sibling embryos at 2 dpf were stained with o-dianisidine staining solution for 15 min in the dark as previously described (*Detrich et al., 1995*). The blood cells from *smarca5$^{zko1049a}$* and their sibling embryos at 2 dpf were collected from heart and caudal vein and attached to slides. The dried slides will be stained by Fast Giemsa Stain (Yeasen Biotech Co., Ltd, CAT: 40,751ES02) following the standard manufacturer's instructions. The K562 cells were collected and washed once using PBS. Then the cells were suspended using 500 µl PBS. Subsequently, add 10 µl 0.4 % benzidine, 1 µl 30 % $H_2O_2$, and 1 µl 5 % sodium nitroferricyanide dihydrate and incubate for 3 min, 5 min and 3 min, respectively. Then the cells were attached to slides for further imaging.

## Chemical treatment

Argatroban (Sigma, A0487), dissolved in DMSO (2 mg/ml), was injected into *smarca5$^{zko1049a}$* and their sibling embryos at 36 hpf at the dosage of 4 nl/embryo. The control embryos were injected with DMSO alone at the same dosage. Heparin (Sigma, H3393), dissolved in $H_2O$ (2.5 mg/ml), was injected into *smarca5$^{zko1049a}$* and their sibling embryos at 36 hpf at the dosage of 4 nl/embryo. For aspirin treatment, the *smarca5$^{zko1049a}$* and sibling embryos at 36 hpf were incubated with aspirin (Sigma, A2093) at the concentration of 5 µg/ml. The *smarca5$^{zko1049a}$* and sibling embryos at 36 hpf were incubated with Glutathione (Sigma, PHR1359) at the concentration of 0.5 mg/ml.

## Confocal microscopy

Confocal microscopy was performed using Nikon confocal A1 laser microscope (Nikon) and Andor high speed confocal (dragonfly, Belfast, UK). The embryos were embedded in 1.2 % low melting agarose.

## Generation of transgenic zebrafish

For overexpression experiment, the full length CDS of *keap1a* was cloned into pDestTol2pA2 with a *hsp70* promoter and an EGFP reporter by DNA assembly (NEBuilder HiFi DNA Assembly Master Mix,

E2621S). The plasmids together with *tol2* mRNA were injected into zebrafish embryos at one-cell stage to generate Tg (*hsp70:flag-keap1a*-EGFP).

## Short interfering RNAs (SiRNAs) and RNA interference

Control and *SMARCA5* siRNAs were synthesized by GenePharma Corporation. The K562 cells were maintained in RPMI-1640 medium supplemented with 10 % FBS and stimulated with hemin (Sigma, 51280, 30 µM) for 3 days to induce erythroid differentiation. Then, the hemin-induced K562 cells were transfected with siRNAs using Lipofectamine RNAiMAX Reagent (Invitrogen, 13778–030) following the manufacturer's instructions. The detailed sequences are listed in *Supplementary file 1C*.

## Western blotting

The western blotting was performed to detect the protein level of SMARCA5 in K562 cells after siRNA transfection. The antibodies used were as followings: anti-Smarca5 antibody (Santa Cruz, H-300: sc-13054), anti-β-Actin antibody (Cell Signaling Technology, 4967).

## Flow cytometry

The *smarca5$^{zko1049a}$* and their sibling embryos with Tg (*gata1*:dsRed) background at 2 dpf were collected and washed by Ringers buffer. After digesting into single-cell suspension using 0.5 % trypsin, the reaction was stopped by adding $CaCl_2$ up to 1 M and fetal calf serum up to 10 %. Then the cells were filtered through 300 Mesh nylon cell-strainer to make single-cell suspension. The RBCs (*gata1*:dsRed$^+$) were sorted using MoFlo XDP (Beckman Coulter) and collected into PBS containing 1 % FBS.

## RNA-seq

RNA-seq was performed in FACS-purified RBCs from *smarca5$^{zko1049a}$* and their siblings at 2 dpf. A total of 50,000 RBCs were used per sample for RNA-seq experiments. The RNAs of sorted HSPCs were isolated using QIAGEN RNeasy Mini Kit (Cat. No. 74104) following the standard manufacturer's instructions. The mRNA libraries were constructed using NEBNext Ultra RNA Library Prep Kit for Illumina and sequenced under Illumina HiSeq X Ten with pair end 150 bp (PE150).

## Processing of RNA-Seq analysis

Raw RNA-seq reads data were trimmed using the fastp (*Chen et al., 2018*) (v2.4) (parameter: with default parameters), and aligned to 'Danio_rerio GRCz10' cDNA reference sequence using the STAR (*Dobin et al., 2013*) (v 2.7.7 a) with the default parameters. Read counts for each gene were quantified as the total number of reads mapping to exons using featureCounts (*Liao et al., 2014*) (subread v1.5.3). DESeq2 (*Love et al., 2014*) was applied to perform differential expression analysis with raw counts quantified by featureCounts. We used Benjamini-Hochberg adjusted p-value < 0.05 and log2 fold change >1 as the threshold for significant difference. Gene set enrichment analysis was performed using *GSEA* function in the clusterProfiler (*Yu et al., 2012*) package (v 3.18.0). Gene set variation analysis was performed by the GSVA (*Hänzelmann et al., 2013*) package (v 1.38.0). The gene sets we used were exported by the msigdbr package (v 7.2.1). The differences in pathway activities scored between *smarca5$^{zko1049a}$* and their sibling RBCs were calculated with limma (*Ritchie et al., 2015*) package (v 3.46.0).

## Assay for transposase-accessible chromatin with high-throughput sequencing (ATAC-Seq)

ATAC-seq was performed in FACS-purified RBCs from *smarca5$^{zko1049a}$* and their siblings at 2 dpf. A total of 50,000 RBCs were used per sample for ATAC-seq library preparation using TruePrep DNA Library Prep Kit V2 for Illumina (Vazyme, TD501) as previously described (*Ding et al., 2021*). Firstly, wash the sorted RBCs using 1xPBST. Then, the cell pellet was lysed using 50 µl cold lysis buffer (10 mM Tris-HCl (pH 7.4), 10 mM NaCl, 3 mM $MgCl_2$ and 0.15% NP-40) for 5 min on ice. Centrifuge and discard the supernatant to get the cell pellet (about 2 µl). Then, the transposition reaction system combining 5xTTBL (10 µl), TTE Mix (5 µl), and $H_2O$ (33 µl) was added immediately to the cell pellet and pipetted up and down gently for several times. After the incubation at 37 °C for 30 min, the DNA was extracted with chloroform-phenol. After the purification, the DNA was amplified using TruePrep DNA Index Kit V2 for Illumina (Vazyme, TD202). After the fragments length purification using VAHTS DNA Clean

Beads (Vazyme, N411), The DNA libraries are under sequencing under Illumina NovaSeq with pair end 150 bp (PE150).

## Processing of ATAC-Seq analysis

Raw ATAC-seq reads were trimmed using cutadapt (v 2.4) (parameter: -q 20 m 20) and mapped to the danRer10 reference genome using Bowtie2 (*Langmead and Salzberg, 2012*) (v 2.3.4.2) (default parameters). Sorting, removal of PCR duplicates and conversion from SAM to BAM files were performed using SAMtools (*Li et al., 2009*) (v 1.3.1). For quality assessment of ATAC-seq libraries, we applied an R package ATACseqQC (*Ou et al., 2018*) (v 1.6.4) to check the fragment size distributions, Transcription Start Site (TSS) enrichment scores, and plot heatmaps for nucleosome positions. We employed deepTools2 (*Ramírez et al., 2016*) (v 2.5.7) to check the reproducibility of the biological replicates and generated bigwig files from BAM output to visualize mapped reads. Peaks were called using MACS2 (*Zhang et al., 2008*) (v2.1.2) (parameter: `--nomodel --nolambda --gsize 1.4e9 --keep-dup` all `--slocal` 10000). Differentially accessible regions were identified using an R package DiffBind (*Ross-Innes et al., 2012*) (v 2.10.0) with a log2 fold change threshold of 0.5, and Benjamini-Hochberg adjusted p-value < 0.1. Peak annotation was performed by an R package ChIPseeker (*Yu et al., 2015*) (v 1.18.0). We identified the enriched de novo motifs across the whole genomic regions using the findMotifsGenome.pl function of HOMER (*Heinz et al., 2010*) (parameter: -size 500 -len 8,10,12 -mask -dumpFasta).

## Transmission electron microscopy

The tail region of *smarca5$^{zko1049a}$* and their siblings at 2 dpf were fixed with 2.5 % (vol/vol) glutaraldehyde and 2 % paraformaldehyde in phosphate buffer (PB) (0.1 M, pH 7.4). After washing with PB for four times, the tissues were immersed in 1 % (wt/vol) $OsO_4$ and 1.5 % (wt/vol) potassium ferricyanide aqueous solution at 4 °C for 1 hr. After washing, the tissues were incubated in filtered 1 % thiocarbohydrazide (TCH) aqueous solution (Sigma-Aldrich) at room temperature for 30 min, followed by 1 % unbuffered $OsO_4$ aqueous solution at 4 °C for 1 hr and 1% UA aqueous solution at room temperature for 2 hr. The tissues were dehydrated through graded alcohol (30%, 50%, 70%, 80%, 90%, 100%, 100%, 10 min each, at 4 °C). Then, transfer the tissues into pure acetone for 10 min (twice). Tissues were infiltrated in graded mixtures of acetone and SPI-PON812 resin (21 ml SPI-PON812, 13 ml DDSA and 11 ml NMA) (3:1, 1:1, 1:3), then transfer the tissues into pure resin. Finally, the tissues were embedded in pure resin with 1.5 % BDMA and polymerized at 45 °C for 12 hr, followed by at 60 °C for 48 hr. The ultrathin sections (70 nm thick) were sectioned with microtome (Leica EM UC6), and examined by a transmission electron microscope (FEI Tecnai Spirit120kV).

## Image analysis

Raw image data were processed using ImageJ, photoshop CC 2018 and Adobe Illustrator CC 2018.

## Statistical analysis

All the statistical analysis was performed for at least three independent biological repeats. GraphPad Prism six was used to analyze the data. Data are mean ± s.d. p Values calculated by two-tailed unpaired Student's *t*-test were used to indicate the significance if not clarified in figure legends.

# Acknowledgements

We thank Lihong Shi and Jun Peng for critical reading of this paper. We are grateful to Xixia Li, and Xueke Tan for helping with electron microscopy sample preparation and taking TEM images at the Center for Biological Imaging (CBI), Institute of Biophysics, Chinese Academy of Science. This work was supported by grants from the National Key Research and Development Program of China (2018YFA0800200), the Strategic Priority Research Program of the Chinese Academy of Sciences, China (XDA16010207), the National Natural Science Foundation of China (31830061, 31425016, and 81530004), and the State Key Laboratory of Membrane Biology, China.

## Additional information

### Funding

| Funder | Grant reference number | Author |
|---|---|---|
| National Key Research and Development Program of China | 2018YFA0800200 | Feng Liu |
| National Natural Science Foundation of China | 31830061 | Feng Liu |
| National Natural Science Foundation of China | 31425016 | Feng Liu |
| National Natural Science Foundation of China | 81530004 | Feng Liu |
| Chinese Academy of Sciences | Strategic Priority Research Program XDA16010207 | Feng Liu |

The funders had no role in study design, data collection and interpretation, or the decision to submit the work for publication.

### Author contributions

Yanyan Ding, Investigation, Methodology, Validation, Visualization, Writing – original draft; Yuzhe Li, Data curation, Methodology; Ziqian Zhao, Investigation, Validation; Qiangfeng Cliff Zhang, Data curation, Supervision; Feng Liu, Conceptualization, Funding acquisition, Project administration, Resources, Supervision, Writing – original draft, Writing – review and editing

### Author ORCIDs

Yanyan Ding (iD) http://orcid.org/0000-0002-3416-2273
Feng Liu (iD) http://orcid.org/0000-0003-3228-0943

### Ethics

The present study was approved by the Ethical Review Committee of the Institute of Zoology, Chinese Academy of Sciences, China (IOZ-IACUC-2020-010).

### Decision letter and Author response

Decision letter https://doi.org/10.7554/eLife.72557.sa1
Author response https://doi.org/10.7554/eLife.72557.sa2

## Additional files

### Supplementary files

• Supplementary file 1. The detailed information of qPCR primers, MOs and siRNAs used in this work. (A) The detailed primers used for qPCR. (B) The detailed sequence and dosage of MOs used in this work. (C) The detailed sequences of siRNAs used in this work.

• Transparent reporting form

### Data availability

The accession number for the sequencing raw data in this paper is BioProject: PRJNA716463. Source data of Figure 6—figure supplement 1D was provided, including the original files of the full raw unedited blots (Figure 6—figure supplement 1—source data 1) and figures with the uncropped blots with the relevant bands clearly labelled (Figure 6—figure supplement 1—source data 2).

The following dataset was generated:

| Author(s) | Year | Dataset title | Dataset URL | Database and Identifier |
|---|---|---|---|---|
| Ding Y, Li Y, Zhao Z, Zhang QC, Liu F | 2021 | Mutation of smarca5 in zebrafish leads to venous thrombosis-like phenotype | https://www.ncbi.nlm.nih.gov/bioproject/PRJNA716463 | NCBI BioProject, PRJNA716463 |

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
