## [Editor Report]

The detailed kinetics and underlying mechanism of thrombosis formation remains elusive. In this study, Liu and colleagues revealed the development of venous thrombosis upon loss of Smarca5 in zebrafish embryos and the Smarca5-mediated epigenetic regulation of oxidative response genes in erythroid cell function. This study not only establishes a new venous thrombosis animal model, but also identify the Smarca5-Keap1-Nrf2-Hmox1 axis as a new regulatory pathway in the pathophysiology of thrombosis.

---

## [Decision Letter]

**Decision letter after peer review:**

Thank you for submitting your article "The chromatin-remodeling enzyme Smarca5 regulates erythrocyte aggregation via Keap1-Nrf2 signaling" for consideration by *eLife*. Your article has been reviewed by 3 peer reviewers, one of whom is a member of our Board of Reviewing Editors, and the evaluation has been overseen by Didier Stainier as the Senior Editor. The following individuals involved in review of your submission have agreed to reveal their identity: Jian Xu (Reviewer #2); Wenqing Zhang (Reviewer #3).

Summary:

All reviewers thought this is an interesting study and most of the experiments are convincingly performed. However, they also raised a number of concerns that need be addressed before we can move forward.

Essential revisions:

1. Figure 3, the electron microscopy studies provide strong evidence for the abnormal organelle morphology including disintegration of mitochondria in Smarca5-deficent RBCs. It would be helpful to provide some quantitative analysis of the size/area and/or number of mitochondria in control and Smarca5-deficent RBCs.

2. Figure 5C, it is somewhat surprising that many of the differentially expressed genes did not show significantly changes in ATAC-seq-based chromatin accessibility. Since the analysis is based on promoter regions, it is possible that many of the differentially expressed genes may be subject to regulation by distal elements such as transcriptional enhancers. It would be helpful to perform additional analysis to include gene-distal ATAC-seq peaks (e.g. +/-100 kb of the TSS) or at least discuss this possibility to explain the lack of overlap between changes in gene expression and ATAC-seq signals.

3. The authors showed that the expression of hmox1a is changed in smarca5 mutants. A variety of genes have been shown to be controlled by Nrf2. More genes should be examined to lend evidence that Nrf2 signaling is indeed perturbed in smarca5 mutants.

*Reviewer #1 (Recommendations for the authors):*

Thrombosis is one of major health issues leading to death worldwide. The venous thrombi are enriched in fibrin and red blood cells (RBCs), which can break off and travel to the lung, causing pulmonary embolism. The detailed kinetics and underlying mechanism of thrombosis formation remains elusive. In this study, Liu and colleagues found that smarca5-deficient red blood cells (RBCs) form blood clots in the caudal vein plexus that mimics venous thrombosis. Mitochondrial cristae are disintegrated in RBCs in smarca5-deficient RBCs. Consistent with that Smarca5 is an epigenetic regulator, loss of smarca5 results in decreased chromatic accessibility at keap1a promoter and a reduction in its expression. Keap1 is a suppressor protein of Nrf2, a master factor controlling the expression of oxidative response genes. The authors further identified that the expression of hmox1a, a downstream target of Nrf2, is markedly increased in smarca5-deficient RBCs. Overexpression of keap1a or knockdown of hmox1a partially rescues the blood clot formation in smarca5 mutants. This study revealed that the chromatin-remodeling enzyme Smarca5 regulates blood clot formation via Keap1-Nrf2 signaling and established smarca5 mutants as a new venous thrombosis animal model. In general, this is an interesting study and most of the experiments are convincingly performed.

*Reviewer #2 (Recommendations for the authors):*

Epigenetic control of gene expression is fundamental for homeostasis such as the development and maintenance of hematopoietic cell lineages. Dr. Liu's group recently described that Smarca5, a major subunit of the SWI/SNF chromatin remodeling complex, is required for the epigenetic programming of fetal HSPCs in zebrafish development (Ding et al., 2021 Blood 137:190-202). However, the roles of Smarca5 in other hematopoietic lineages are largely known. In this manuscript, the authors first described that the Smarca5-deficient (smarca5-zko1049a) zebrafish embryos tend to form venous blood clots. The Smarca5-deficient RBCs displayed disintegration of cristae in mitochondria by electron microscopy. Using comprehensive chromatin profiling (by ATAC-seq) and transcriptomic analysis (by RNA-seq), they identified Keap1, a suppressor of Nrf2, as one of the downstream targets of Smarca5 in RBCs. Altered Keap1 and Nrf2 expression resulted in dysregulation of oxidative response genes, whereas Keap1a overexpression or depletion of the Nrf2 target gene Hmox1a can partially rescued the blood clot formation in Smarca5 mutants.

Overall, this is an important and well-executed study describing multiple new findings related to the development of venous thrombosis upon loss of Smarca5 in zebrafish embryos and the Smarca5-mediated epigenetic regulation of oxidative response genes. The detailed characterization of Smarca5-deficient zebrafish by various genetic models, live imaging and anti-thrombosis drug treatment were well designed and executed, and the results support the main conclusions. A major strength of the current study is the rigorous dissection of the Smarca5-deficient zebrafish model using orthogonal genetic, pharmacologic and rescue approaches. The integrative analysis of multiple data types and the inference of the regulatory pathways therein are well performed. This study not only establishes a new venous thrombosis animal model, but also identify the Smarca5-Keap1-Nrf2 axis as a new regulatory pathway in the pathophysiology of thrombosis. Thus, this work will be of broad interest to the study of erythropoiesis, thrombosis, and regulation of oxidative signaling pathways.*Reviewer #3 (Recommendations for the authors):*

In this study, the authors identified the role of Smarca5 in red blood cell (RBC) aggregation in zebrafish caudal vein plexus. They further profiled the chromatin accessibility landscape and transcriptome features in RBCs and found that the disrupted Keap1-Nrf2 signaling is responsible for the RBC aggregation phenotypes in smarca5 mutants.

Overall, this work characterizes a novel role for smarca5 in RBC aggregation and performed detailed analysis by confocal imaging and high-throughput sequencing. However, whether this phenotype is indeed the thrombosis phenotype requires further consideration. We suggest that the authors could describe this phenotype as RBC aggregation but not thrombosis throughout the whole text.

---

## [Author Response]

Essential revisions:1. Figure 3, the electron microscopy studies provide strong evidence for the abnormal organelle morphology including disintegration of mitochondria in Smarca5-deficent RBCs. It would be helpful to provide some quantitative analysis of the size/area and/or number of mitochondria in control and Smarca5-deficent RBCs.

Thanks for this comment. We have performed the quantitative analysis of the area and number of mitochondria in control and *smarca5*-deficent RBCs. The results showed that the area of mitochondria was not significantly changed and the number of mitochondria was slightly increased but not significantly changed in *smarca5*-deficent RBCs. These data were added in Figure 3F and G in the revision.

2. Figure 5C, it is somewhat surprising that many of the differentially expressed genes did not show significantly changes in ATAC-seq-based chromatin accessibility. Since the analysis is based on promoter regions, it is possible that many of the differentially expressed genes may be subject to regulation by distal elements such as transcriptional enhancers. It would be helpful to perform additional analysis to include gene-distal ATAC-seq peaks (e.g. +/-100 kb of the TSS) or at least discuss this possibility to explain the lack of overlap between changes in gene expression and ATAC-seq signals.

We greatly appreciate the reviewer’s inspiring comment and the helpful guidance. We agree with the reviewer that it is possible that many of the differentially expressed genes may be subject to regulation by distal elements such as transcriptional enhancers. We therefore followed the reviewer’s guidance, and examined the genes in which the chromatin accessibility at distal regions and their transcription were both increased or decreased after *smarca5* deletion (see Figure 5-figure supplement 1E). We found that the overlap between changes in gene expression and ATAC-seq signals increased when taking the distal regions into consideration, indicating that some of the differentially expressed genes may be subject to the regulation by distal ATAC-seq peaks.

We think the possible reasons for the lack of overlap between changes in gene expression and ATAC-seq signals are as follows.

First, gene expression is regulated by a variety of regulatory factors, such as trans-regulatory elements and cis-regulatory elements (Wittkopp, Haerum, and Clark, 2004). In general, complex interactions between cis-regulatory elements and trans-regulatory elements control gene expression (Gibson and Weir, 2005; Hill, Vande Zande, and Wittkopp, 2021; Wittkopp, 2005). However, the peak annotation strategy is based on the distance from peak to the TSS (transcriptional start site) of its nearest gene (Yu, Wang, and He, 2015), which may lead to a situation that the peak is annotated as gene A because it is closest to the gene A but it actually regulates gene B expression. If the peak is differentially accessible and only gene B is differentially expressed, we are not able to find the overlap between changes in gene B expression and the peak (annotated as gene A) signal.

Second, cells exhibit signiﬁcant variations in gene expression and the underlying regulation of chromatin because of intrinsic and extrinsic factors (Ma et al., 2020). A recent study that applies single cell multi-omics sequencing (SHARE-seq) found that during lineage commitment, chromatin accessibility at domains of regulatory chromatin (DORCs) precedes gene expression, indicating that changes in chromatin accessibility may prime cells for lineage commitment (Ma et al., 2020). DORCs were defined as high-density peak-gene-associated regions.

The authors systematically analyzed the cuticle/cortex trajectory and revealed that DORCs generally become accessible prior to onset of their associated genes’ expression. For example, they observed sequential activation of peaks in the Wnt3 DORC, with individual enhancer peaks activating much earlier than the Wnt3 promoter, followed by activation of nascent RNA expression (estimated by intron counts) and, ﬁnally, mature RNA expression (estimated by exon counts). Therefore, the accessibility of peaks and the expression of genes are not exactly matched, which may contribute to explaining the lack of overlap between changes in gene expression and ATAC-seq signals. We have added the related discussion about these possibilities in the revision (Page 20-21, line 365-373, in red).

3. The authors showed that the expression of hmox1a is changed in smarca5 mutants. A variety of genes have been shown to be controlled by Nrf2. More genes should be examined to lend evidence that Nrf2 signaling is indeed perturbed in smarca5 mutants.

Thanks for this thoughtful comment. Besides *hmox1a*, *gclc*, *ggt1b*, *gsr*, *gstp1*and *gstk1*, more target genes of Nrf2, including *fbp1a*, *gsto2*, *prdx1*, *pgd* and *g6pd* were further examined in RBCs from *smarca5^zko1049a^* and their siblings. The results showed that most of these genes were perturbed in RBCs after *smarca5* deletion, indicting the perturbed Nrf2 signaling in *smarca5* mutants.